

# A study of the influence of tropospheric subsidence on spring and summer surface ozone concentrations at the JRC-Ispra station in northern Italy

Pavlos Kalabokas[1], Niels Roland Jensen[2], Mauro Roveri[3], Jens Hjorth[2,4], Maxim Eremenko[5], Juan Cuesta[5], Gaëlle Dufour[5], Gilles Foret[5] and Matthias Beekmann[5]

[1]Academy of Athens, Research Center for Atmospheric Physics and Climatology, Athens, Greece.
[2]European Commission, Joint Research Centre (JRC), Directorate for Energy, Transport and Climate, Air and Climate Unit, I-21027 Ispra (VA), Italy.
[3]European Commission, Joint Research Centre (JRC), Directorate for Nuclear Safety and Security, I-21027 Ispra (VA), Italy.
[4]Department of Environmental Science, iCLIMATE, Aarhus University, Frederiksborgvej 399, 4000 Roskilde, Denmark
[5]Laboratoire Interuniversitaire des Systèmes Atmosphériques (LISA), UMR7583, CNRS, Université Paris-Est-Créteil, Université de Paris, Institut Pierre Simon Laplace, Créteil, France.

*Correspondence to*: Pavlos Kalabokas (pkalabokas@academyofathens.gr)

**Abstract.** The influence of tropospheric ozone to the surface ozone concentrations is investigated at the monitoring station of JRC-Ispra, based on 10 years of measurements (2006-2015) of surface ozone data. In-situ hourly measurements of ozone and other air-pollutants, meteorological parameters and weekly averaged $^7$Be (as indicator of upper-tropospheric/stratospheric influence) and $^{210}$Pb measurements (as indicator of boundary layer influence), have been used for the analysis. It is observed that frequently $^7$Be and ozone weekly peaks coincide, which might be explained by the impact of deep atmospheric subsidence on surface ozone, particularly during late spring and early summer. Based on this observation, a detailed analysis of selected $^7$Be and ozone episodes occurring during that period of the year has been performed in order to further elucidate the mechanisms of tropospheric influence to the surface pollutant concentrations. For the analysis composite NOAA/ESRL reanalysis synoptic meteorological charts in the troposphere have been used as well as IASI satellite ozone measurements and NOAA-HYSPLIT back trajectories. The JRC-station hourly measurements during subsidence episodes show very low values of local pollution parameters (e.g. $NO_x$, $^{222}$Rn, nephelometer data, $PM_{10}$), close to zero. On the contrary, during these periods ozone levels reach usually values around 45-60 ppb during the afternoon hours but also show significantly higher values than the average during the night and morning hours, which is a sign of direct tropospheric influence to the surface ozone concentrations.

## 1 Introduction

It has been reported that tropospheric ozone levels as well as surface ozone concentrations have increased significantly during the last couple of centuries (Volz and Kley, 1988; Forster et al., 2007). Ozone is an important greenhouse gas and



might cause adverse effects to human health and also have negative impacts on vegetation and materials (Ito et al., 2005; Van Dingenen et al., 2009; Hollaway et al., 2012). Tropospheric ozone is mainly produced in the troposphere through photochemical reactions of precursor pollutants but it does also originate from stratospheric intrusions (Volz and Kley, 1988; Staehelin et al., 1994). On the average, the tropospheric ozone originating from the stratosphere is about 10 % on global scale (Monks et al., 2015). The surface ozone concentrations depend on photochemistry and transport within the boundary layer as well as tropospheric entrainment, which might also be associated to deep tropospheric subsidence especially over the Mediterranean during summer where, in general, high ozone levels are observed. (Lelieveld et al., 2002; Zerefos et al., 2002; Kalabokas et al., 2007; Kalabokas et al., 2008; Lelieveld et al., 2009; Velchev et al., 2011, Richards et al., 2013; Kalabokas et al., 2013; Cooper et al., 2014; Safieddine et al., 2014; Kalabokas et al., 2015).

The influence of intercontinental ozone transport on surface ozone concentrations is considered as a critical issue regarding ozone pollution control (UNESCE, 2010) and this issue has been extensively studied especially in the USA where many studies have shown that tropospheric transport and entrainment of ozone from the free troposphere to the boundary layer has an important impact on surface ozone-mixing ratios (Cooper et al., 2011; Cooper et al., 2012; Parrish et al., 2013; Parrish et al., 2014; Langford et al., 2015). Also, anticyclonic synoptic conditions are normally linked to tropospheric subsidence, which is considered to be an important source of high ozone concentrations over the eastern Mediterranean. During the summer period, the Mediterranean area is under the influence of the descending branch of the Hadley circulation (Lelieveld, 2009) in combination with the impact of the Indian monsoon, inducing a Rossby wave that through the interaction with the mid-latitude westerlies produces adiabatic descent in the area (Rodwell and Hoskins, 1996, 2001; Tyrlis et al., 2013). In the eastern Mediterranean strong deep subsidence throughout the troposphere to the boundary layer, which seems to be a quite frequent phenomenon, has been documented based on the analysis of MOZAIC vertical ozone profiles as well as surface ozone and satellite measurements (Kalabokas et al., 2007, 2008, 2013, 2015; Eremenko et al., 2008; Foret et al., 2009; Liu et al., 2009; Doche et al., 2014; Gaudel et al., 2018). In addition, large-scale atmospheric modeling simulations (Li et al., 2001; Richards et al., 2013; Zanis et al., 2014; Safieddine et al., 2014; Tyrlis et al., 2014; Akritidis et al., 2016; Cristofanelli et al., 2018) show the importance of the vertical ozone transport over the Mediterranean basin, especially in its eastern side. Also, sea-breeze circulation appears to have a particularly strong influence on ozone formation in the western Mediterranean, because it favors accumulation of ozone in recirculated polluted air masses (Millan et al. 1997, 2000; Castell et al. 2008). The atmospheric processes controlling ozone levels over the western and central Mediterranean need further studying, especially for what concerns the springtime months. It has been reported that surface background ozone levels in the western and central Mediterranean show a maximum in spring (April–May) while in the eastern Mediterranean stations a later ozone maximum appears in July–August (Kalabokas et al., 2008; Zanis et al., 2014). Also, it has been recently reported that high spring-time ozone concentrations are detected over a large geographical area for several days under specific synoptic conditions, which could be explained by the impacts of tropospheric subsidence (Kalabokas et al., 2017).



Atmospheric radionuclides are useful for studies of tropospheric subsidence and transport (WMO-GAW, 2004; Feichter et al., 1991; Balkanski et al., 1993; Koch et al., 1996; Graustein and Turekian, 1996; Arimoto et al., 1999; Gerasopoulos et al., 2001; Zanis et al., 2003; Liu et al., 2004; Cuevas et al., 2013). Due to their different origins, the investigation of $^{210}$Pb and $^{7}$Be activities simultaneously can be useful for studies of atmospheric transport of pollutants, especially in particle phase

(Koch et al., 1996). $^{210}$Pb has a half-life of about 22 years. It originates from the decomposition of $^{222}$Rn, which has a half-life if 3.8 days and it is a decomposition product of $^{226}$Ra originating from the ground (Baskaran, 2011). $^{7}$Be has a cosmogenic origin and it is formed mostly by the decomposition of the atoms of carbon, nitrogen and oxygen present in the atmosphere by incident gamma radiation. It has a half-life of 53.3 days (Masarik and Beer, 1999). Its production rate increases with altitude and saturates at about 15 km height (Usokin and Kovaltsov, 2008). Most of $^{7}$Be is produced in the

stratosphere and about the one third in the troposphere, especially in its upper part (O'Brien, 1979). After their formation, $^{7}$Be atoms are mostly attached to atmospheric particles and so their atmospheric concentrations are greatly influenced by transport and deposition processes of particles (Papastefanou and Ioannidou, 1995; Jaenicke, 1988; Koch et al., 1996; Feely et al., 1989).

On the average, the $^{7}$Be concentrations in the upper troposphere are about the 25% of the lower stratospheric concentrations (about160 mBq m$^{-3}$) at northern mid-latitudes while $^{7}$Be concentrations close to the ground are generally below 5 mBq m$^{-3}$ (Reiter et al., 1983; Dutkiewicz and Husain, 1985; Brost et al., 1991; Gaggeler, 1995). So, air masses originating from the upper troposphere or stratosphere contain usually high $^{7}$Be concentrations and the intrusions of stratospheric air mass into the troposphere are the main processes transferring $^{7}$Be to the earth's surface through dry or wet deposition. Therefore, $^{7}$Be

serves as a tracer for lower stratospheric and upper tropospheric air masses arriving to the ground (Lee et al., 2007; Papastefanou et al., 2012). Due to the exchange and removal processes in the atmosphere the $^{7}$Be concentration in air at ground level varies strongly with the season while $^{7}$Be production rates in the atmosphere remain relatively constant (Durana et al., 1996; Masarik and Beer, 1999). In general, $^{7}$Be surface concentrations show a maximum in late summer (Reiter et al., 1983; Feely et al., 1989).

The investigation of the atmospheric processes controlling the frequently observed springtime ozone maximum over parts of the European continent including the western Mediterranean is an interesting research issue as photochemical ozone production and tropospheric transport, including stratospheric influence, might be involved (Beekmann et al., 1994; Monks, 2000). It has been previously reported that high surface ozone concentrations, lasting several days, have been observed over

large geographical areas at the same time (Kalabokas et al., 2017). It has been also shown that the observed regional springtime ozone episodes are usually associated to specific synoptic meteorological patterns, which have great similarities with those observed  during summertime ozone episodes over the eastern Mediterranean and linked to large-scale subsidence (Kalabokas et al., 2013, 2015; Doche et al., 2014).



The present study focuses on the influence of tropospheric subsidence on surface ozone concentrations, especially during spring and summer months over the western Mediterranean area. In extension to previous work, it is based on the analysis of 10 years of ozone and other air pollution measurement data (2006-2015) as well as measurements of natural radionuclide tracers at surface level, which can be used as tracers of transport and photochemical and removal processes. These data are

collected at the JRC-Ispra site, located in the pre-alpine area in northern Italy but also located relatively close to the western Mediterranean. In addition, we used meteorological maps, back trajectories and IASI satellite ozone data for a better understanding of the relative importance of the contributions of the different sources of ozone, especially the role of the vertical transport in the troposphere. To our knowledge, a similar analysis of such multi-parameter long term measurements has not yet been performed so far for this area.

## 2 Experimental

### 2.1 Site description

The JRC-Ispra station (45.807°N, 8.631°E, 223m a.s.l) is located in a valley in the pre-alpine area of northern Italy. A general meteorological characteristic of the area is that low winds usually prevail with occasional northerly Foehn events (Mira-Salama et al., 2008). More details about the site can be found in Putaud et al., 2017.

### 2.2 Instrumentations and measurements at JRC-Ispra site

The following instrumentation has been used during this investigation:

Ozone (Thermo 49 instruments, UV Photometric Ambient Analyzer). For the data analysis the mid-day (12:00 – 18:00) mean values were used as most representative of boundary layer air.

$^7$Be and $^{210}$Pb measurements, as weekly averages. For more details see in Jensen et al., 2017.

$^{222}$Rn activity concentrations (ANSTO dual-flow loop two-filter detector). Hourly averages have been used. For more details

on the method see in Zahorowski et al., 2004.

$NO_x$ (NO, $NO_2$) Thermo 42/42iTL instrumentations (Chemiluminescent Nitrogen Oxides Analyser). Hourly averages have been used. . For more details on the method see in Villena et al., 2012.

Nephelometer (Particle Scattering and Backscattering Coefficient, Instrument TSI 3563). Hourly averages have been used. For more details about the instrument, see Putaud et al., 2017.

$PM_{10,}$ TEOM (Tapered Element Oscillating Mass balance, Series 1400a). Hourly averages have been used. For more details about the instrument, see Putaud et al., 2017.

Regarding the meteorological measurements, a WXT510 weather transmitter from "Vaisala" recorded simultaneously 6 meteorological parameters, temperature, pressure, relative humidity, precipitation, wind speed and wind direction, from the top of an about 10 m high mast. Humidity (in ppmV units) was calculated from relative humidity (RH), temperature and



atmospheric pressure, which was useful for the data-analysis as an air mass reaching the surface by transport from higher altitudes will typically be relatively dry and entrainment of air from the free troposphere is thus normally associated with a drop in humidity mixing ratio as well as specific humidity.

## 2.3 Meteorological maps and back trajectories

Composite NOAA/ESRL reanalysis meteorological charts for various meteorological parameters for selected ozone episodes and for the atmospheric pressure levels of 850, 700 and 500 hPa have been produced with a horizontal resolution of 2.5° × 2.5° (Kalnay et al., 1996). Due to space limitations mostly the 700 hPa charts, representative for the free troposphere, are presented. In addition, 6-day NOAA /HYSPLIT back trajectories for selected high ozone days for air masses arriving at the JRC-Ispra site at various end points have been plotted, using the GDAS meteorological data (Draxler and Rolph, 2015).

## 2.4 IASI and IASI+GOME2 satellite ozone measurements

Progresses in satellite observations of tropospheric ozone have been made during the last decade (e.g., Fishman et al., 2003; Liu et al., 2005; Coheur et al., 2005; Worden et al., 2007; Eremenko et al., 2008, Cuesta et al., 2013). These progresses combined with their spatial coverage and horizontal resolution make possible to use such observations to complement in situ observations and to support the analysis of ground measurements as well as modeling simulations. In this study, we use ozone satellite observations derived from the IASI infrared instruments and from the multispectral synergism of IASI and GOME-2 in the ultraviolet, for enhancing sensitivity to ozone closer to the surface. The first IASI instrument (Clerbaux et al., 2009) has been launched on board the MetOp-A platform on 19 October 2006. It is a Fourier transform spectrometer operating at nadir in the thermal infrared between 645 and 2760 cm$^{-1}$ with an apodized spectral resolution of 0.5 cm$^{-1}$. IASI monitor atmospheric composition twice a day at any (cloud-free) location at high resolution with its swath width of 2200 km and its field of view composed of a 2×2 pixels matrix with a diameter at nadir of 12 km each (e.g., Boynard et al., 2009; George et al., 2009; Clarisse et al., 2011; Coman et al., 2012). As in Doche et al. (2014), IASI ozone concentrations retrieved at 3 km and 10 km height are used for our analysis, as representative of the lower and upper troposphere, respectively. The maximum of sensitivity of IASI retrievals ranges between 3 and 5 km in the lower troposphere and 9 and 12 km in the upper-troposphere. Several studies show that the ozone concentrations retrieved from IASI in the lower- and the upper-troposphere are mainly uncorrelated (Dufour et al., 2010, 2012, 2015). We use the vertical profiles retrieved from IASI to calculate longitudinal transect for different latitudes. Vertical profiles within 1° in latitude and 0.5° in longitude are averaged in the transect calculation.





The satellite multispectral approach used here is called IASI+GOME2. It is based on the joint and simultaneous use of both GOME-2 and IASI measurements for deriving a unique ozone profile for each pair of spectra. It is designed for observing lowermost tropospheric ozone located below 3 km of altitude (with typically a peak of maximum sensitivity down to 2 km of altitude), which is not directly observed with single-band retrievals. As IASI, GOME-2 is only onboard the MetOp satellite

series and offer global coverage every day (for MetOp-A around 09:30 local time) with a swath width similar to IASI and a ground resolution moderately coarser than IASI (pixels of 80 km × 40 km for GOME-2). As described in detail by Cuesta et al., (2013), co-located IR and UV spectra are jointly fitted to retrieve a single vertical profile of ozone for each pixel at the IASI horizontal resolution. The UV measurements from the closest GOME-2 pixel (without averaging) are used for each IASI pixel. As for IASI only retrievals, a priori ozone profiles representative of tropical, mid-latitude and polar conditions

are calculated by averaging the climatological ozone profiles from McPeters et al., (2007) over the 20-30°N, 30-60°N and 60-90°N latitude bands. The selection of the a priori profiles used during the retrieval is based on the tropopause heights (determined by the temperature vertical profile for each IASI pixel) above 14 km, between 14 and 9 km and below 9 km, respectively. IASI+GOME2 retrievals are routinely produced at the global scale by the French data centre AERIS and they are publicly available (see https://www.aeris-data.fr, http://cds-espri.ipsl.fr and https://iasi.aeris-data.fr).

## 3 Results and discussion

### 3.1 Seasonal variation of $O_3$ concentrations, $^7Be$ concentrations and the $^7Be/^{210}Pb$ ratio.

For the investigation of entrainment episodes, plots of weekly averages of ozone vs. $^7Be$ concentrations and of ozone vs. $^7Be/^{210}Pb$ ratios were made for all years during the examined period (2006-2015) and the year 2011 is shown as an example (Figs. 1a, 1b). As observed, $^7Be$ and $O_3$ peaks are in several cases coinciding, which might be explained by an impact of deep

atmospheric subsidence (air masses moving down from the stratosphere/upper troposphere) on surface ozone. Particularly during springtime, the high ozone levels during such events may be influenced by ozone rich air being transported down to the boundary layer from the stratosphere/upper troposphere.

As seen in Figure 1, in May 2011 and in a period around end of June 2011 and beginning of July 2011 there were episodes of

downward transport of ozone down to the surface from the above tropospheric layers. The relatively higher $^7Be/^{210}Pb$ ratios at the end of the spring to the beginning of the summer seasons (mid-April to mid-July) indicate that stratospheric or upper tropospheric influence should be most important during this period.

In addition, in Figure 2 the $^7Be$ activity and the $^7Be/^{210}Pb$ ratio are presented together with specific humidity. As observed,

very often the local $^7Be/^{210}Pb$ maxima coincide with local minima of specific humidity, which supports the assumption, that this isotope ratio is an indicator of the relative importance of entrainment of subsiding dry air originating from the upper atmospheric into the boundary layer and the ground surface. The $^7Be$ activity does not show a similar correlation with





specific humidity. If we look at the yearly variation of the isotope ratio, it has a maximum in the early summer while specific humidity has a maximum later in the summer, which may be explained by the fact that warm air can contain more water vapor.

The $^7Be/^{210}Pb$ peaks are in some cases coinciding with ozone peaks but local $^7Be$ activity peaks are found to be more frequently coinciding with local ozone maxima. Thus the radioisotope data are consistent with the hypothesis that maximum ozone values are frequently reached in situations where there is a combined impact of entrainment of ozone rich air brought down by subsidence (high $^7Be$ activities) and stagnant atmospheric conditions (high $^{210}Pb$ activities), favoring ozone formation in the boundary layer. In such conditions, $^7Be$ activities will be high, but not the $^7Be/^{210}Pb$ ratio. This is illustrated
in Figures 1 and 2 for the year 2011, but a similar picture is seen during other years, where observations are available (see Supplement Figs S1-S4).

As reported in the literature, stratospheric ozone intrusions in the Mediterranean occur most frequently from March to July (Beekmann et al., 1994; Monks et al., 2000). An ozone increase is generally observed during spring months, associated with
the increase in solar radiation and photochemistry (Monks et al., 2000) but also from tropospheric downward transport (Kalabokas et al., 2017) and thus some caution would be needed in the data-analysis as the increase in $O_3$, $^7Be$ and $^7Be/^{210}Pb$ at the same time is not necessarily related to the impact of stratospheric intrusions. However, the occurrence of $^7Be/^{210}Pb$ peaks above the baseline level can be interpreted as evidence of deep atmospheric subsidence followed by entrainment into the boundary layer. The episodes in May and June 2011 with high ozone and $^7Be/^{210}Pb$ values (Figs 1, 2) appears to be due
to such deep tropospheric subsidence as indicated by synoptic meteorological maps, back-trajectories and IASI satellite data. Such specific episodes will be examined in detail later in the following section.

Table 1 shows the average monthly values for spring, summer and autumn (March to October) of $^7Be$, $^7Be/^{210}Pb$ and $O_3$ (12-18 afternoon mean) for all of the years 2006-2015 where measurements were available. The $^7Be$ concentrations show
maximum monthly values in June-July, same as ozone, while the $^7Be/^{210}Pb$ ratio has its peak values in the April – June period which would indicate enhanced deep tropospheric subsidence during this period of the year.

**3.2 Investigation of the main synoptic characteristics during the May- July ozone episodes**

As the May-July period seems to be the most important one regarding the influence of subsidence on ozone concentrations, as a next step selected high $^7Be$ and ozone weekly episodes during the May-July period will be examined. 10 $^7Be$ - ozone weekly episodes were examined and the 3 most characteristic of them, out of more than 10 present in the whole time series will be presented in the following paragraphs. The episodes discussed here are not Foehn events.





In Figure 3 the composite NOAA/ESRL weather maps of geopotential height, wind speed, omega vertical velocity and specific humidity for the episode of 3-10 May 2011 (as well as the corresponding 3-10 May climatology charts) at JRC-Ispra are shown. A strong low-pressure system is observed over the Atlantic and a weaker one over Eastern Europe and the Black-Sea while anticyclonic conditions prevail over Western and Central Europe in a vast region extending from Scandinavia to the Mediterranean. In this region strong subsidence is observed as deduced from the charts of omega vertical velocity and specific humidity, indicating large-scale downward movement of dry air masses. Similar subsidence conditions have been already observed during regional spring ozone episodes over the western Mediterranean, especially related to the interface of an anticyclone and a low-pressure system located further east (Kalabokas et al., 2017). The above feature is also clearly seen in the following selected episode of 23-29 May 2012 (Fig. 4), with a much stronger anticyclone established over the North Sea (weekly averages measured at Ispra: $^{7}Be$: 7117 $\mu Bq\ m^{-3}$, $^{7}Be/^{210}Pb$ ratio: 9.6, Ozone 54.9 ppb). The corresponding IASI and IASI+GOME2 satellite images as well as IASI vertical sections at various latitudes from 60-45º N for ozone concentration at 3 km and 10 km are shown in Figs. 5-8. The IASI+GOME2 maps show enhanced ozone concentrations at 3km over Germany, France and Italy, which seem disconnected from the values at 10 km. In a general way, IASI-GOME2 ozone concentrations at 3km seem more independent from those at 10 km, than are those derived from IASI at the same altitudes, which is expected because the number of degrees of freedom is larger. It appears that the geographical distribution of tropospheric ozone as well as the movement of the high ozone reservoirs at both altitude levels generally follow the synoptic weather patterns (Figs. 4, 5). An extended high ozone area appears to the east of the anticyclone at both tropospheric levels apparently originating from the north, being an extension of the northern polar high-ozone reservoir. It has to be added, that back-trajectories show air masses arriving to the Ispra site from N-NE directions and from higher altitudes, especially on May 25-27 (not shown). This behaviour is usually encountered in the analysis of many spring ozone episodes over the area as the 6-day back-trajectories usually originate from the region of high tropospheric ozone subsidence over central and eastern Europe, thus inducing high ozone background levels of tropospheric origin at the boundary layer and at the surface to which the photochemical ozone build-up might eventually be added.

The next episode of 28 June – 04 July 2011 seems to be quite representative of early summer ozone episodes over the area and will be examined in more details by taking into account many relevant atmospheric measurements recorded at the JRC-Ispra station. During this episodic event, very high $^{7}Be$ concentrations as well as $^{7}Be/^{210}Pb$ ratio values have been recorded, which were actually the highest weekly averages for 2011 (Figs. 1a,b). Regarding the weather conditions, it has been mostly sunny throughout the week with some rain on the second day and light north-westerly winds.

The synoptic conditions (Figs 9-10, Fig. S5) show the existence of an extended deep low-pressure system over the Atlantic in the free troposphere (at 500 -700hPa), and also another one over central Europe and the boundary layer (at 700-1000hPa), while high atmospheric pressure prevails over most of the European continent, including Scandinavia. Following the



examination of the meteorological charts, it has been observed that massive subsidence occurs over a wide area over the Atlantic and western Europe towards the Mediterranean, including the Alpine region, peaking on 1-3 July. Indeed, specific humidity charts at various pressure levels (Fig. S5) show an extended area of dry air masses over the Atlantic (N. Spain – W. France) at lower latitudes, moving towards the European continent and the Mediterranean following the synoptic flow (Fig. 9). The omega vertical velocity charts show that the descending motion is stronger at higher altitudes but at the same time there is an accumulation of dry air masses over the Atlantic (indicating subsidence), which are displaced according to the above described synoptic pattern. In addition, the tropospheric ozone distribution as measured by the IASI and IASI+GOME2 satellites at 3 and 10 km follows the synoptic patterns (Figs. 11-14).  As observed on June 30 and July 1, there is a large zone of enhanced ozone at 10 km, but also at 3 km, descending from the North Sea to the Alps, and which corresponds to a through east of the ridge.  During July 2 and 3, the through has developed to a cut-off low located  over SE Poland, and so do high ozone values at 10 km.

The above described characteristics related to subsidence are much more pronounced if only 1-2 July is considered, especially the large-scale descending tendency of tropospheric air masses at all pressure levels over Central and Northern Europe, which is subsequently associated with extended tropospheric dryness over these areas (Figs S6-S8).

In addition, according to the back-trajectory plots the subsiding air masses arrive to the JRC-Ispra site from the Atlantic coast and France (Fig. 16). Almost all back-trajectories arrive from the north, where the subsidence area is located (Figs 9-10, Fig. S5) and where a large area of tropospheric ozone appears over Western Europe, apparently originating from the high ozone reservoir located over polar regions as shown in the corresponding IASI and IASI+GOME2 satellite images as well as IASI vertical sections (Figs 11-14).

The hourly air pollution measurements at the JRC-station during the 28 June – 04 July 2011 period show first a period (June 28 and 29) with large ozone (daily maximum more than 80 ppb) and $PM_{10}$ (daily maximum more than 80 µg/m³), related to high temperatures ($T_{max}$ 30°C and low wind speed (<1 m/s). Aerosol is of mainly anthropogenic origin as indicated by different nephelometer responses to red, green and blue light. High ozone values during this period are probably due mainly to photochemical build-up from anthropogenic emissions in the Po valley.  During the following period with maximum of subsidence (1-2 July), ozone concentrations vary around 55-60 ppb with the diurnal concentration range significantly reduced in comparison to the previous days while at the same time the $NO_x$ concentrations get minimum values (Fig. 13) as well as the humidity, $^{222}Rn$, nephelometer and $PM_{10}$ concentrations (Fig. 16-18). Please note that the weekly resolution of $^7Be$ measurements does not allow for ascertaining an expected maximum during these two days. During the days following the strongest subsidence period (3-4 July) the nocturnal ozone values are significantly reduced as the tropospheric entrainment has diminished while the ozone destruction by NO chemical titration and dry deposition on the ground reduces substantially the concentrations within the generally shallow nocturnal layer. On the other hand, the mid-day ozone





concentrations are slightly increased probably due to in-situ photochemical ozone production, which is added-up to the increased tropospheric ozone background, due to the regional tropospheric subsidence episode occurring during the previous days. Thus, this period shows an interesting suite of days with strong photochemical ozone production and advection of tropospheric ozone to the ground.

In summarizing the above, it has to be mentioned at first that maximum ozone values in the area of the study are expected to be connected primarily with boundary layer ozone photochemistry. The presented analysis shows that regional tropospheric subsidence occurring frequently in the area during May – July might enable easier exceedances of ozone air quality standards, as photochemical ozone build-up is initiated in a clean boundary layer air mass containing already high ozone

levels (i.e. 50 ppb), which is common after a regional tropospheric subsidence episode in the area. This phenomenon may explain the spring ozone maximum observed over many stations over the Western and Central part of the European continent including the Western Mediterranean. In relation to the above, it has been mentioned in the introduction that a similar phenomenon is observed in the Eastern Mediterranean, maximizing though later in summer (July – August).

## 4 Conclusion

High boundary layer concentrations of $^7$Be are used as indicator of the influence of free tropospheric air, in which $^7$Be activity is high due a cosmogenic source. $^{210}$Pb activity is an indicator of accumulation of surface emissions and their reaction products and it reaches its highest levels during periods with stagnant air conditions. Radioisotope and humidity data from the Ispra station show that local peaks in $^7$Be/$^{210}$Pb frequently are coinciding with local minima in specific humidity, consistent with the hypothesis that these peaks are found in situations with a strong impact of tropospheric

influence to the boundary layer. Comparison with ozone measurements shows that these peaks in some occasions coincide with ozone peaks but more frequently ozone peaks coincide with peaks in $^7$Be activity. This observation was interpreted as a result of the fact that the highest ozone concentrations frequently are found in situations with a combined impact of entrainment of ozone rich free tropospheric air and local formation in the boundary layer. The conclusions derived from the analysis of specific episodes were in accordance with this interpretation: The main characteristics of the frequently occurring

spring episodes where both $^7$Be and ozone reach maxima at the Ispra station were found to be the following:

* Anticyclonic stagnant conditions over parts of the European continent and the western Mediterranean,

* Strong winds at the periphery of the anticyclone associated with a deep low-pressure system located to the North and a weaker one located to the East. A common feature is that Ispra is located at the eastern edge of a ridge system at 700 hPa level and at the same time a trough is located eastwards,

* Very extended areas of positive vertical velocity, omega (downward movement), observed over eastern, central and western Europe, depending on the locations and the relative strength of the high and low pressure systems, at all pressure levels and associated with dry conditions (low specific humidity), indicating subsidence.



At the same time, the IASI satellite images show important ozone reservoirs in the upper and lower troposphere, which are generally delimited by the meteorological systems and follow their movement while large areas of enhanced tropospheric ozone appear over the region of subsidence, usually originating from the tropospheric ozone reservoirs associated with the low-pressure systems. These results consolidate the findings of the first phase of this study on spring ozone episodes in the

western Mediterranean (Kalabokas et al., ACP, 2017), and extend them over a full year and a longer time period. The characteristics described above are also encountered during some summer episodes (in June-July) but the conditions generally observed in summer episodes are more related to local photochemical ozone production in the boundary layer while tropospheric subsidence is weaker and more concentrated over the Eastern Mediterranean. The examination of the Ispra station hourly measurements during subsidence episodes shows that the local pollution parameters (e.g. $NO_x$, $^{222}Rn$,

nephelometer, $PM_{10}$) tend to have low values (as compared to those observed during periods of anthropogenic pollution), while the ozone levels usually reach values around 45-60 ppb during the afternoon hours but show significantly higher values than the average during the night and morning hours. This is a clear sign of tropospheric entrainment to the boundary layer. During high $^7Be$ and high ozone episodes, the highest evening ozone values exceeding the standards usually occur within the following 2-3 days after the maximum of regional tropospheric subsidence, as observed also in the analysis of

several episodes not presented in this paper, usually under the influence of favourable meteorological conditions for photochemical ozone production in the boundary layer, which is added-up on the increased regional background due to tropospheric subsidence and thus occasionally leading to exceedances in ozone air quality standards. The results of this study might be useful for helping the required improvements in the veracity of global ozone air quality models for which biases have been made evident by several recent studies (Cooper et al., 2014; Parrish et al., 2014; Gaudel et al., 2018; Young et al.,

20   2018).

**Acknowledgement**

The authors would like to acknowledge the following people: S. Dos Santos and J.-P. Putaud for meteorological data and

aerosol physics data. G. Manca for $^{222}Rn$ data. F. Lagler and J.-P. Putaud for the help with the ozone and $NO_x$ data. The authors also acknowledge that a small part of the material in the present paper has been used in ref. Jensen et al., 2017 (Figure 1). Acknowledgement is also made for the composite weather maps, which were provided by the NOAA/ESRL Physical Sciences Division, Boulder, Colorado, from their website at http://www.cdc.noaa.gov/. The authors also acknowledge the NOAA Air Resources Laboratory (ARL) for the provision of the HYSPLIT transport and dispersion model

and/or READY website (http://www.ready.noaa.gov) used in this publication. LISA acknowledges the support from CNES (Centre National des Etudes Spatiales)/TOSCA (Terre Océan Surface Continentale Atmosphère), and PNTS (Programme National de Télédétection Spatiale) for the development and production of ozone observations from IASI+GOME2 and IASI.





**Author contributions**

PK, NRJ and JH prepared the manuscript with contributions from all co-authors to the manuscript/data-evaluation. MR provided $^7$Be and $^{210}$Pb data. NRJ and JH provided ozone and $NO_x$ data. PK provided synoptic meteorological maps and back-trajectories. GF, GD, ME and MB provided the IASI satellite ozone measurements.

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



**Table 1: Monthly averages of $^7$Be, $^{210}$Pb, ozone (12:00 – 18:00) concentrations and $^7$Be/ $^{210}$Pb ratio during the March – October period (2006-2015).**

| Month | $^7$Be ($\mu$Bq m$^{-3}$) | $^{210}$Pb ($\mu$Bq m$^{-3}$) | $^7$Be/$^{210}$Pb ratio | O$_3$ 12-18 (ppb) |
|---|---|---|---|---|
| 3 | 4031.2 | 875.3 | 6.1 | 25.0 |
| 4 | 4737.1 | 835.4 | 7.1 | 34.9 |
| 5 | 4741.0 | 726.7 | 7.3 | 40.2 |
| 6 | 4863.8 | 794.4 | 7.0 | 52.6 |
| 7 | 5824.5 | 1049.7 | 6.6 | 52.5 |
| 8 | 5272.3 | 1114.8 | 5.7 | 36.5 |
| 9 | 4490.8 | 1090.8 | 4.7 | 25.4 |
| 10 | 4249.3 | 1609.7 | 3.6 | 16.3 |




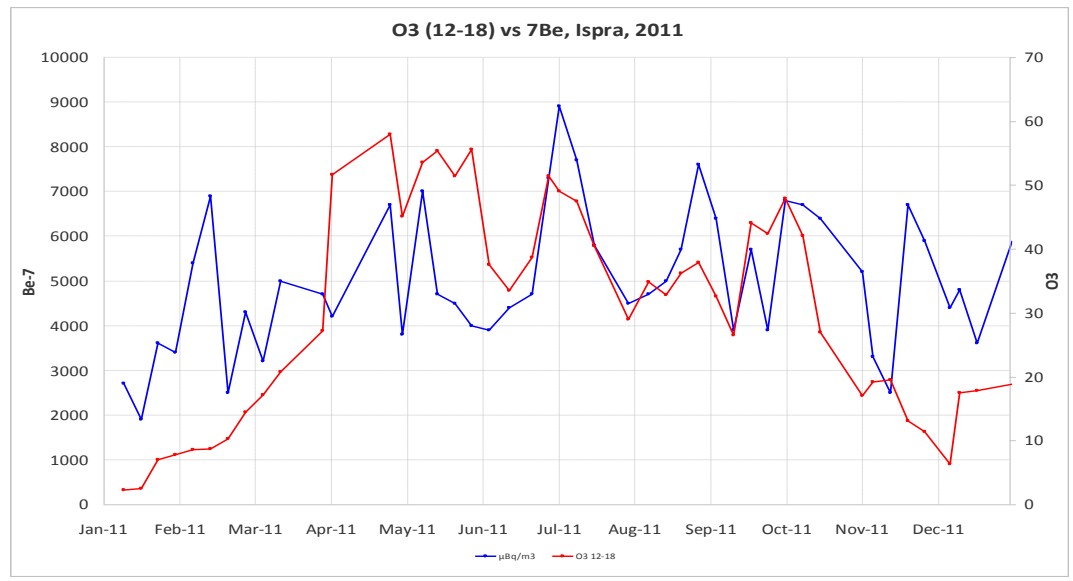

**Figure 1a: Weekly averages for ozone 12:00-18:00 (ppb, red) and $^{7}$Be (mBq m$^{-3}$, blue) at the JRC-Ispra station for 2011. The authors acknowledge the use of a similar figure in ref. Jensen et al., 2017 (see the 'acknowledgements' section for more details).**

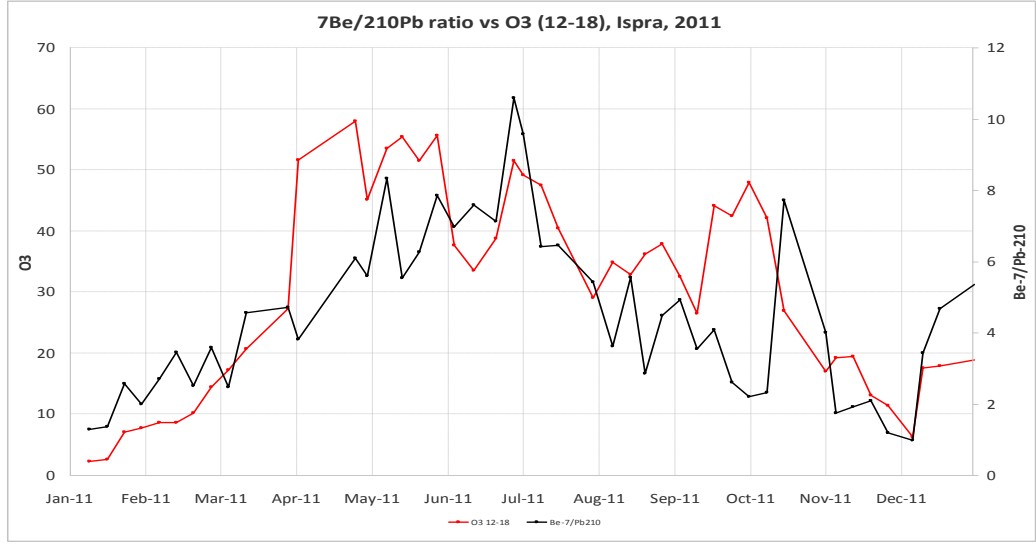

**Figure 1b: Weekly averages for ozone 12:00-18:00 (ppb, red) and $^{7}$Be/$^{210}$Pb ratio (black) at the JRC-Ispra station for 2011. The authors acknowledge the use of a similar figure in ref. Jensen et al., 2017 (see the 'acknowledgements' section for more details).**




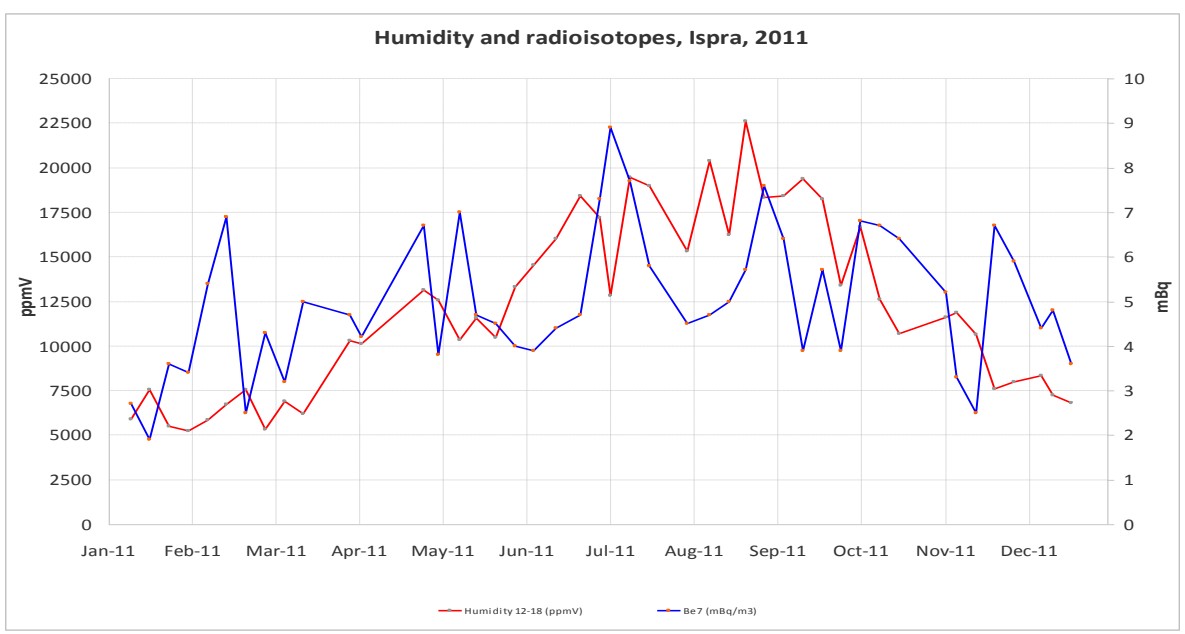

**Figure 2a: Weekly averages of the ⁷Be concentrations (mBq m⁻³, red) and specific humidity 12:00-18:00 (ppmV, blue) at the JRC-Ispra station for 2011.**

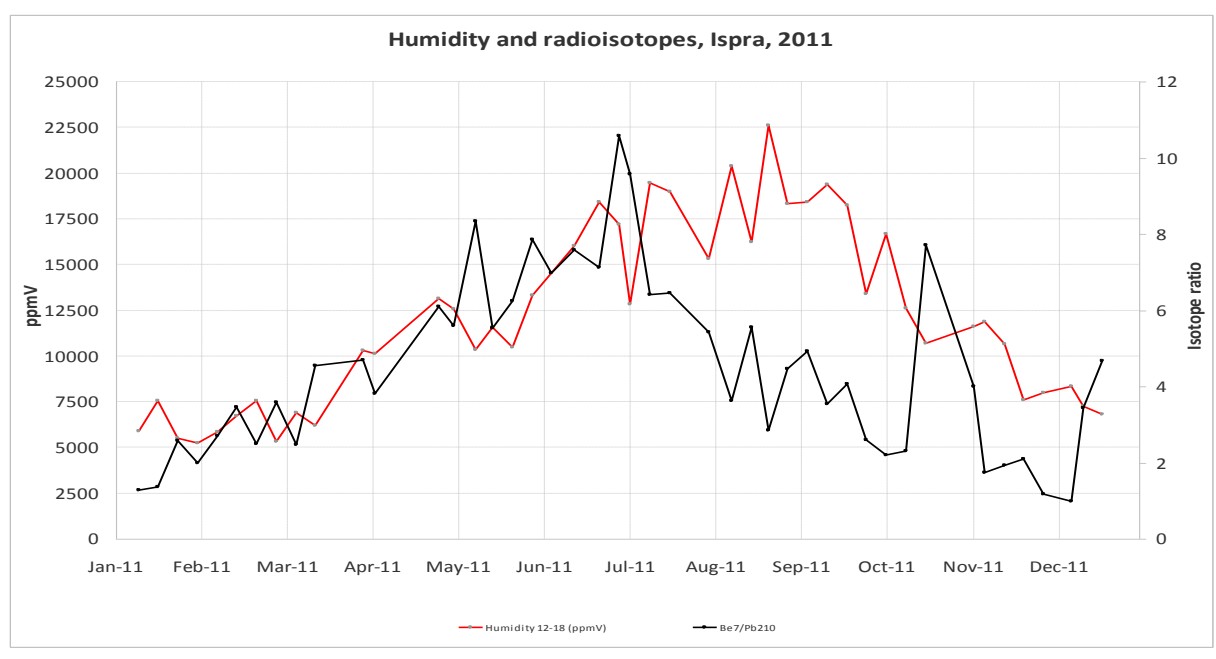

**Figure 2b: Weekly averages of the ⁷Be /²¹⁰Pb ratio and specific humidity 12:00-18:00 (ppmV, blue)at the JRC-Ispra station for 2011.**



5 **Figure 3: Composite NOAA/ESRL weather maps of geopotential height, vector wind speed, omega vertical velocity and specific humidity for 3-10 May climatology (left column) and for the episode of 3-10 May 2011 at JRC-Ispra, Italy (right column).**





**Figure 4: Composite NOAA/ESRL weather maps of geopotential height, vector wind speed, omega vertical velocity and specific humidity for 23-29 May climatology (left column) and for the episode of 23-29 May 2012 at JRC-Ispra, Italy (right column).**





**Fig. 5: IASI satellite images for ozone concentration at 3 km (left column) and 10 km (right column) for the episode of 23-29 May 2012. Starting from the top: 23-29 May, 23-24 May, 25-27 May, 28-29 May.**



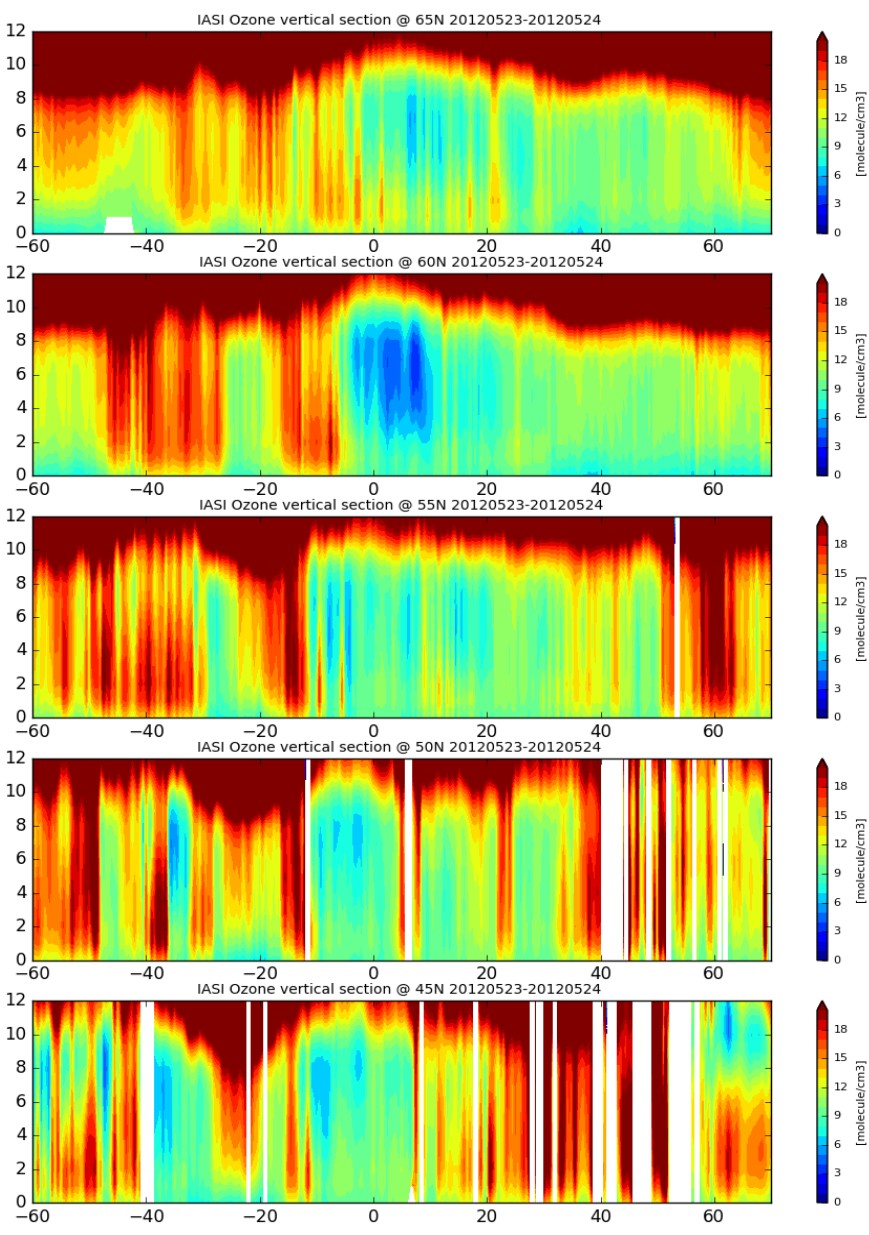

**Fig. 6: IASI vertical sections at various latitudes from 65º N (upper panel) to 45 º N (lower panel) on May 23-24, 2012.**



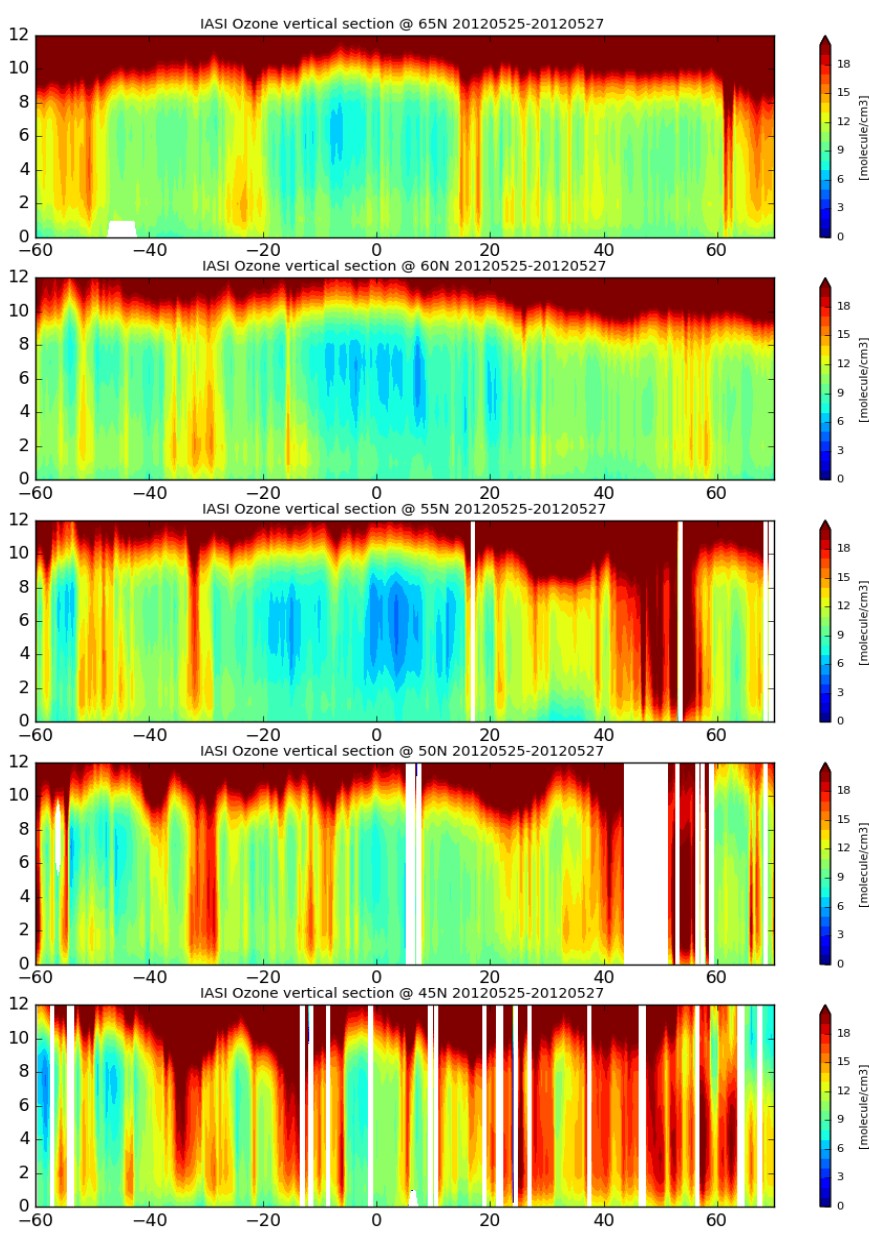

**Fig. 7: IASI vertical sections at various latitudes from 65º N (upper panel) to 45º N (lower panel) on May 25-27, 2012.**

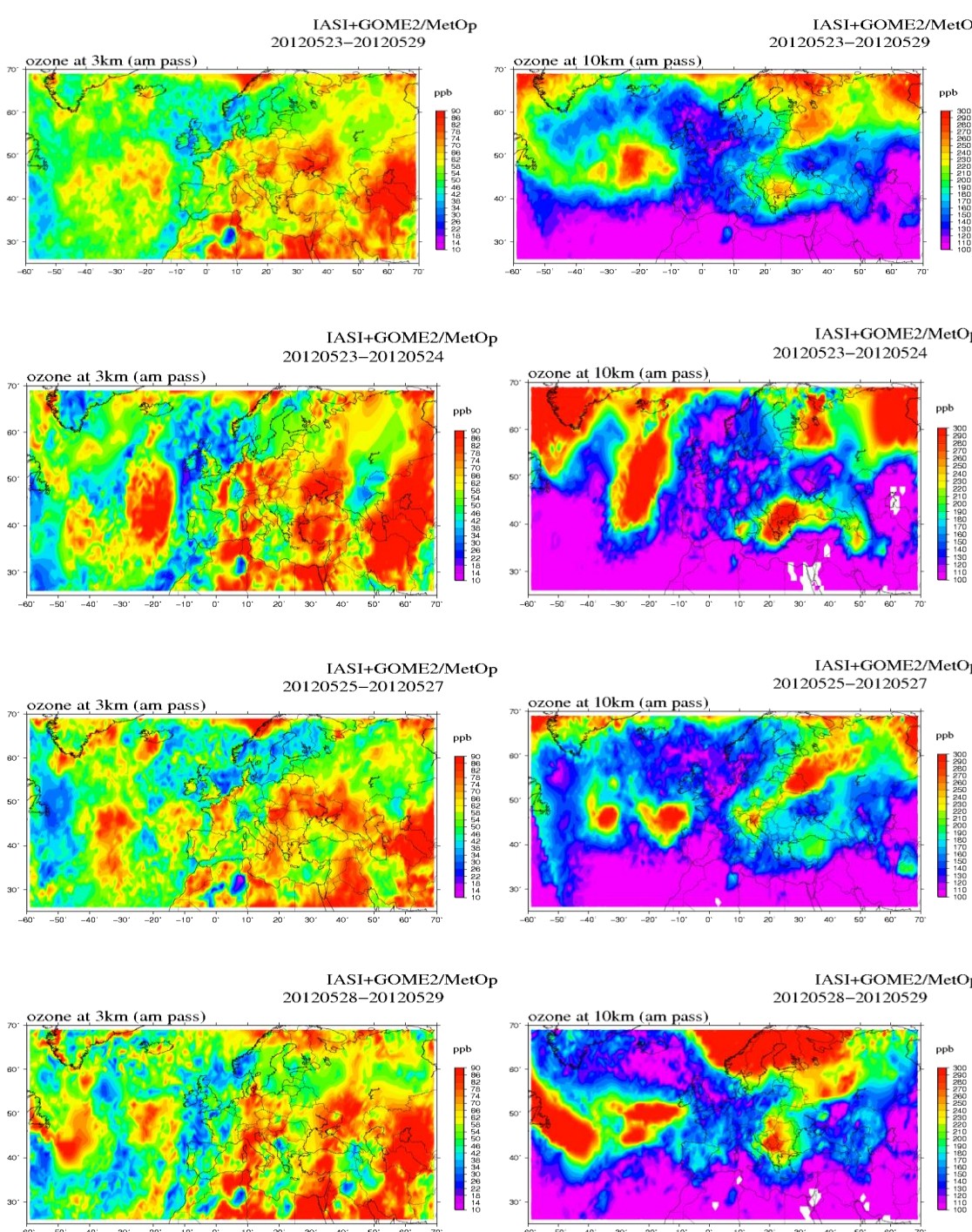

5  **Fig. 8: IASI +GOME2 satellite images for ozone concentration at 3 km (left column) and 10 km (right column) for the episode of 23-29 May 2012. Starting from the top: 23-29 May, 23-24 May, 25-27 May, 28-29 May.**



5   **Figure 9: Composite NOAA/ESRL weather maps of geopotential height, vector wind speed, omega vertical velocity and specific humidity for 28 June-04 July 2011 climatology (left column) and for the episode of 28 June-04 July 2011 at JRC-Ispra, Italy (right column).**





5 **Fig. 10: (Left column, from above): Composite charts for specific humidity anomaly at 850 hPa 5-days, 3-days, 2-days before 28 June-04 July 2011 and on 28 June-04 July 2011 (lower panel).**

**(Right column, from above): Composite charts for specific humidity anomaly at 1000 hPa 5-days, 3-days, 2-days before  28 June-04 July 2011 and on 28 June-04 July 2011 (lower panel).**





**Fig. 11: IASI satellite images of ozone concentration at 3 km (left column) and 10 km (right column) for the episode of 28 June – 04 July 2011.  Starting from the top: 28 June – 04 July , 30 June – 01 July,  02 July – 03 July.**



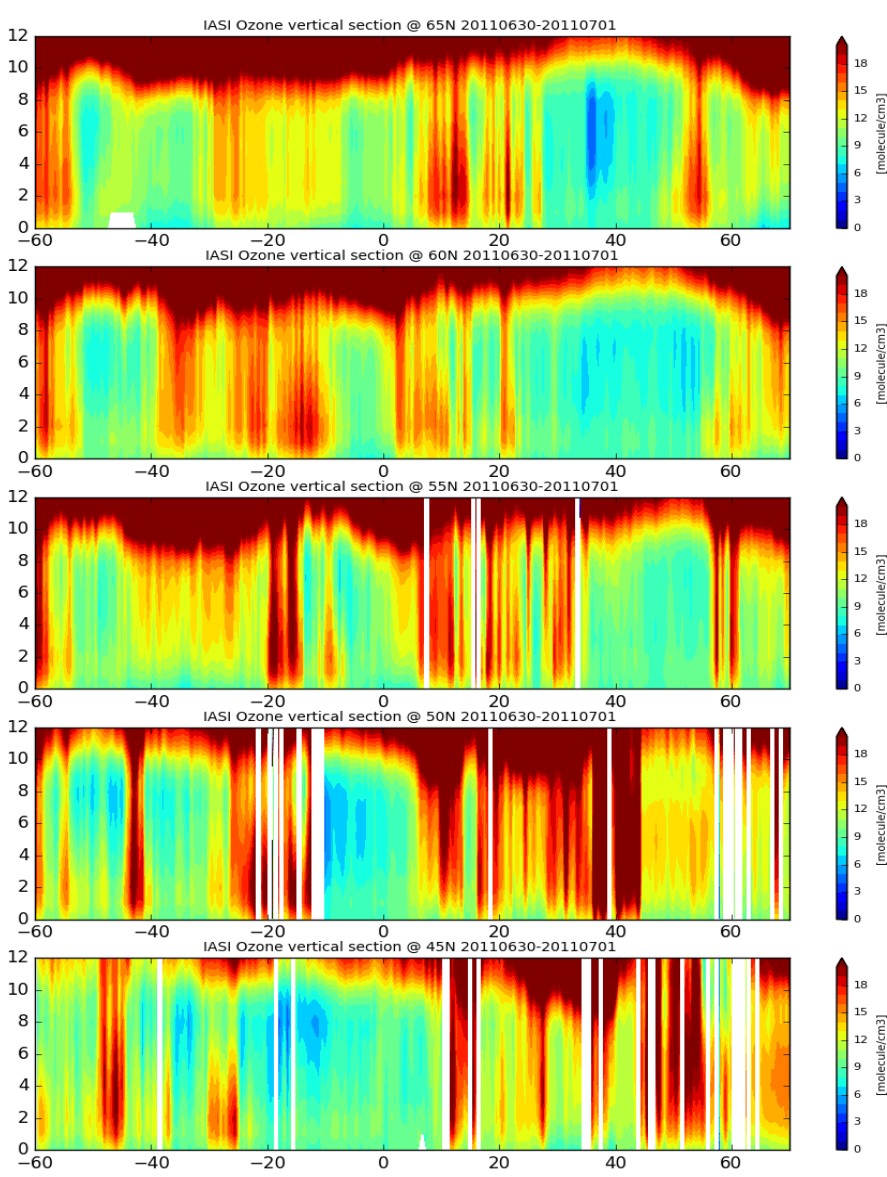

**Fig. 12: IASI vertical sections at various latitudes from 65 ºN (upper panel) to 45 N (lower panel) on June 30-July 1, 2011.**




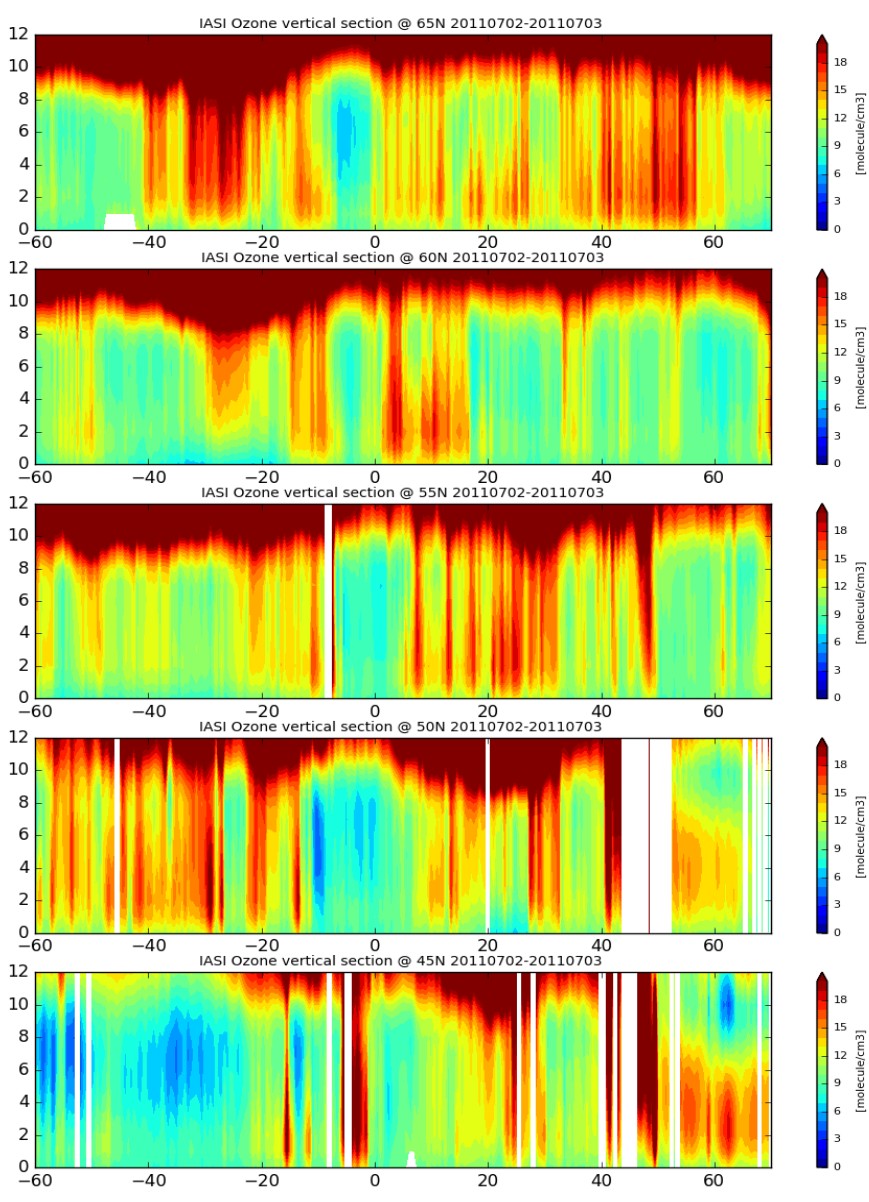

**Fig. 13: IASI vertical sections at various latitudes from 65 °N (upper panel) to 45 °N (lower panel) on July 2-3, 2011.**



**Fig. 14: IASI+GOME2 satellite images of ozone concentration at 3 km (left column) and 10 km (right column) for the episode of 28 June – 04 July 2011. Starting from the top: 28 June – 04 July, 30 June – 01 July, 02 July – 03 July.**



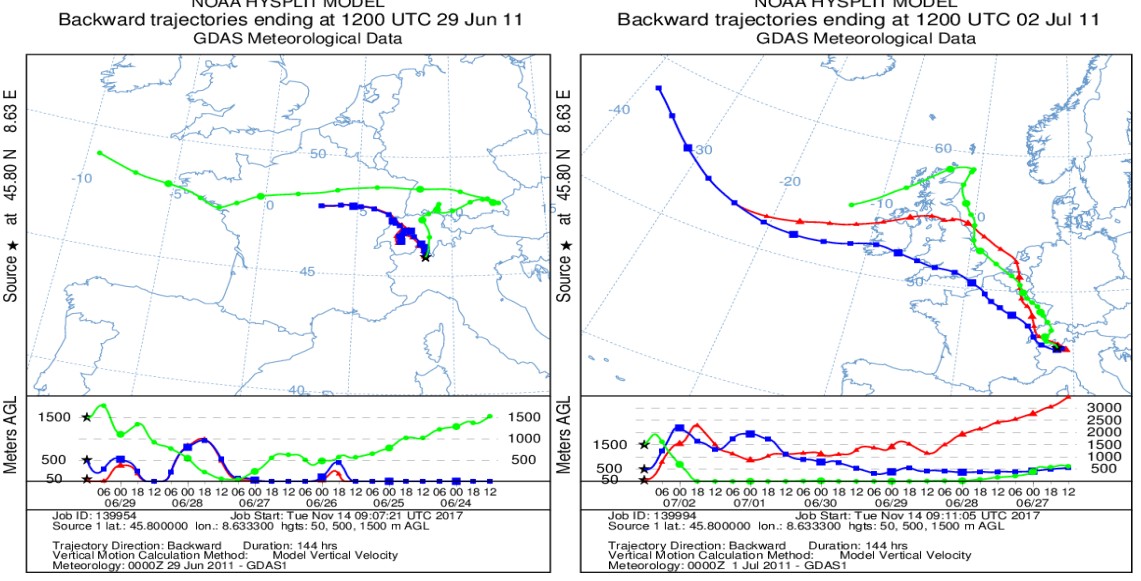

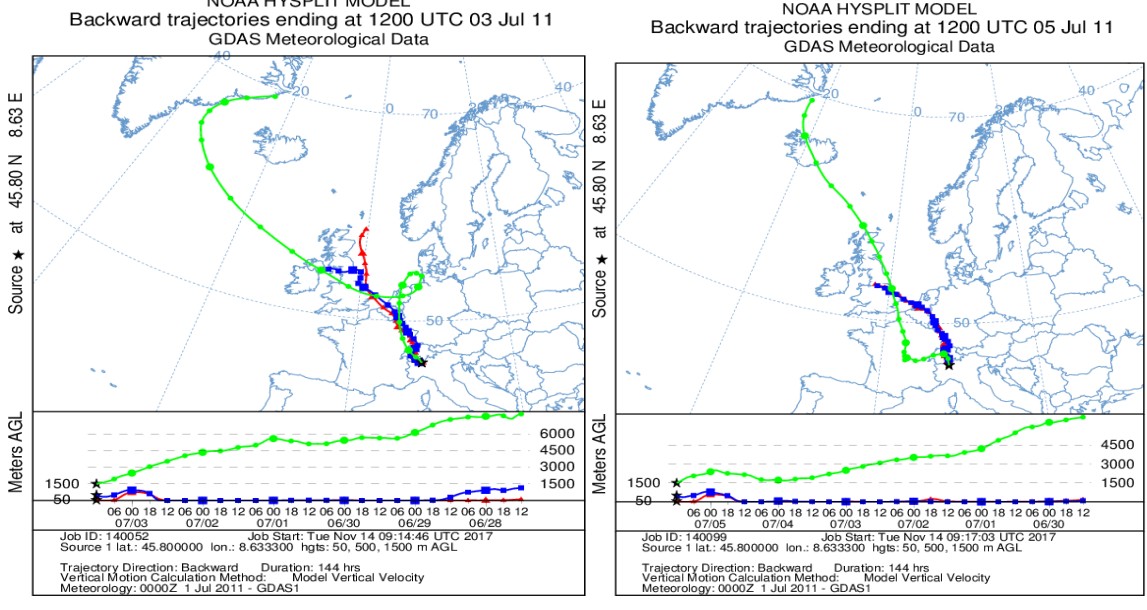

**Fig. 15: 6-day HYSPLIT back-trajectories arriving at the JRC-Ispra station during the episode of 28 June-04 July 2011.**



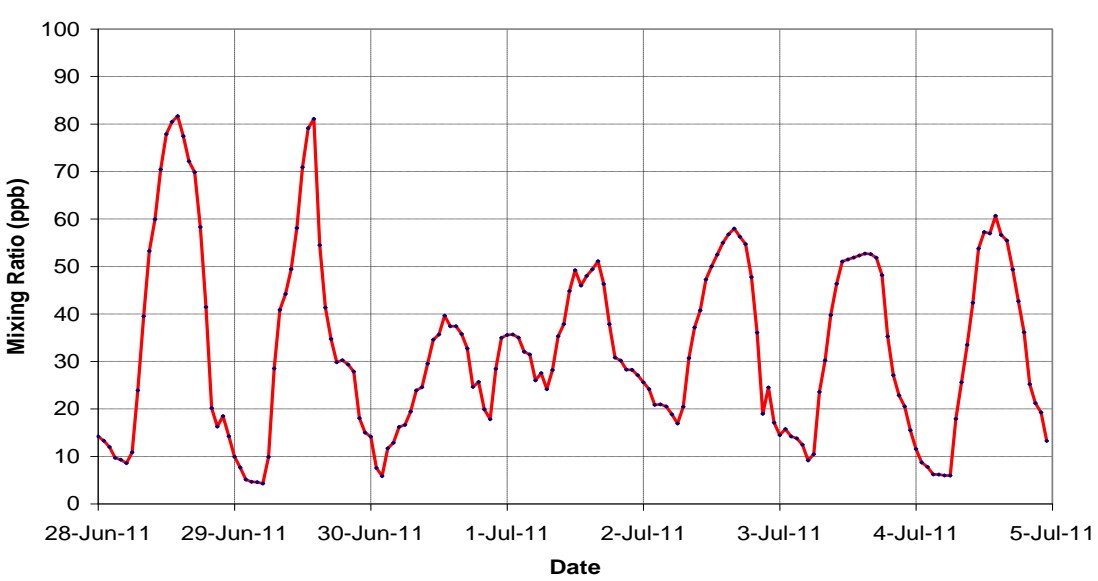

**Fig. 16a: Hourly ozone concentrations for 28 June-04 July 2011 at JRC-Ispra, Italy.**

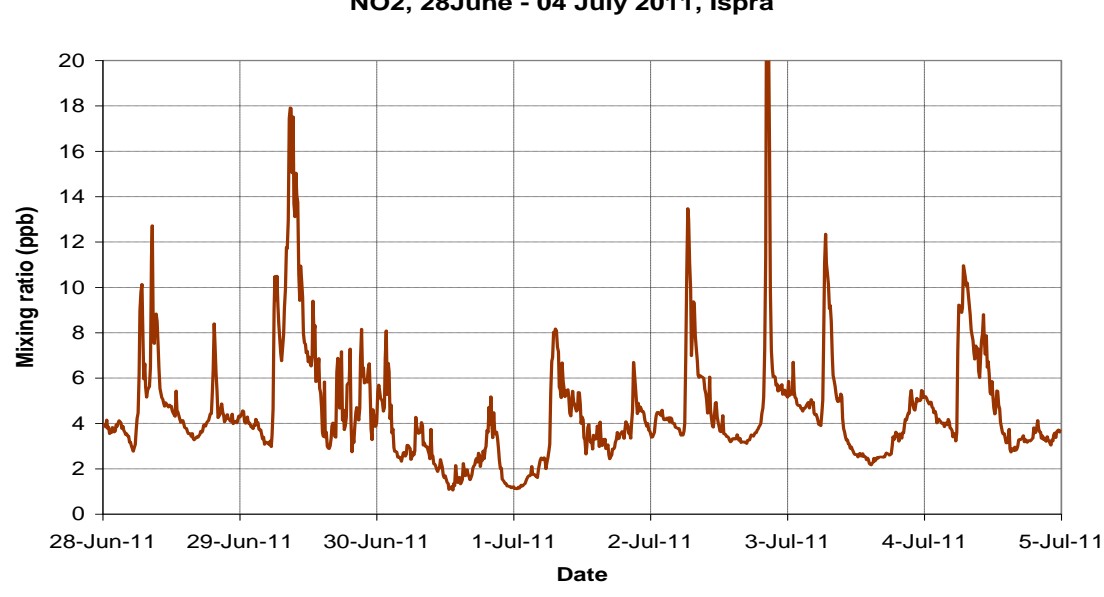

**Fig. 16b: Hourly nitrogen dioxide concentrations for 28 June-04 July 2011 at JRC-Ispra, Italy.**





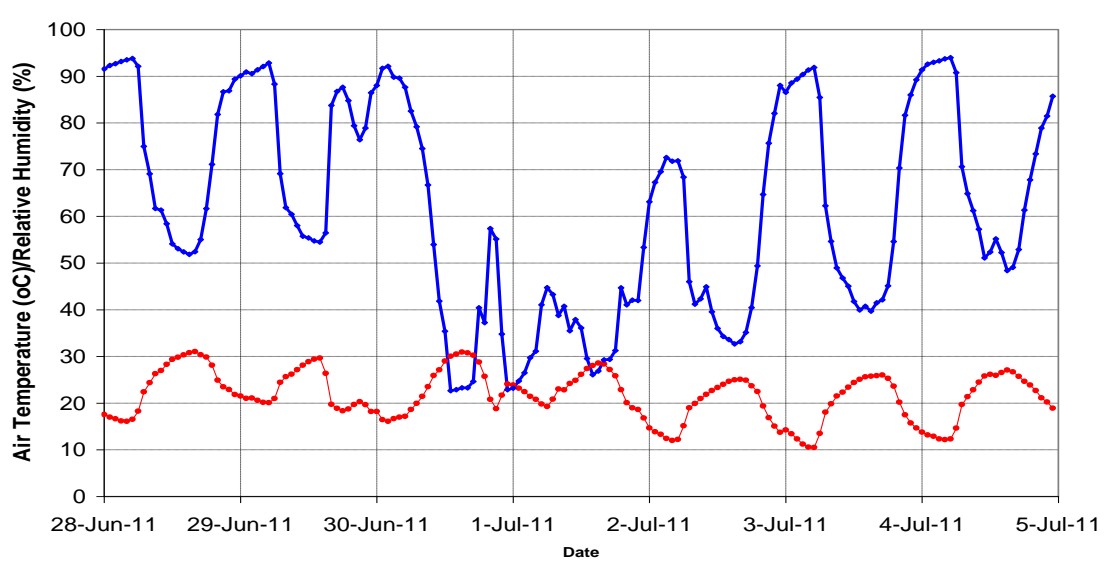

**Fig. 17a: Hourly Relative Humidity (blue) and temperature (red) measurements for 28 June-04 July 2011 at JRC-Ispra, Italy.**

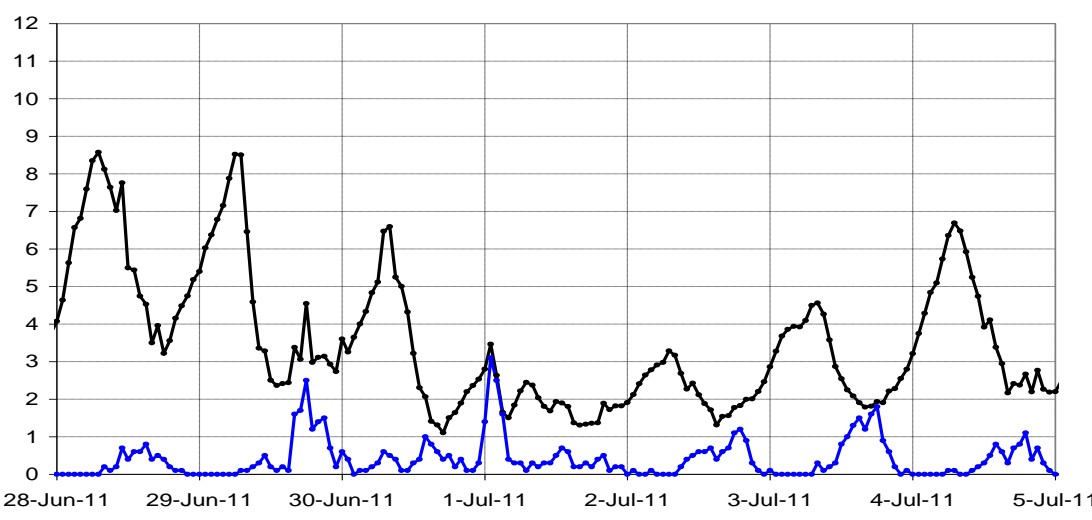

**Fig. 17b: Hourly $^{222}$Rn concentrations (in Bq m$^{-3}$, black) and Wind Speed (in m s$^{-1}$, blue) for 28 June-04 July 2011 at JRC-Ispra, Italy.**



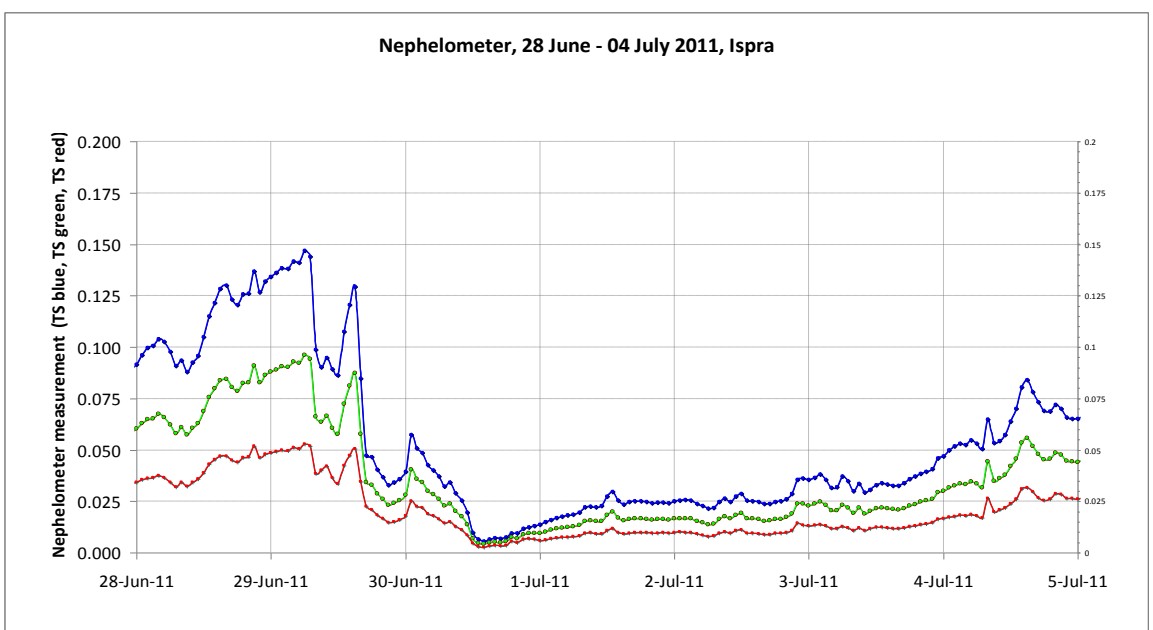

**Fig. 18a: Hourly Nephelometer measurements (in km$^{-1}$), for 28 June-04 July 2011 at JRC-Ispra, Italy.**

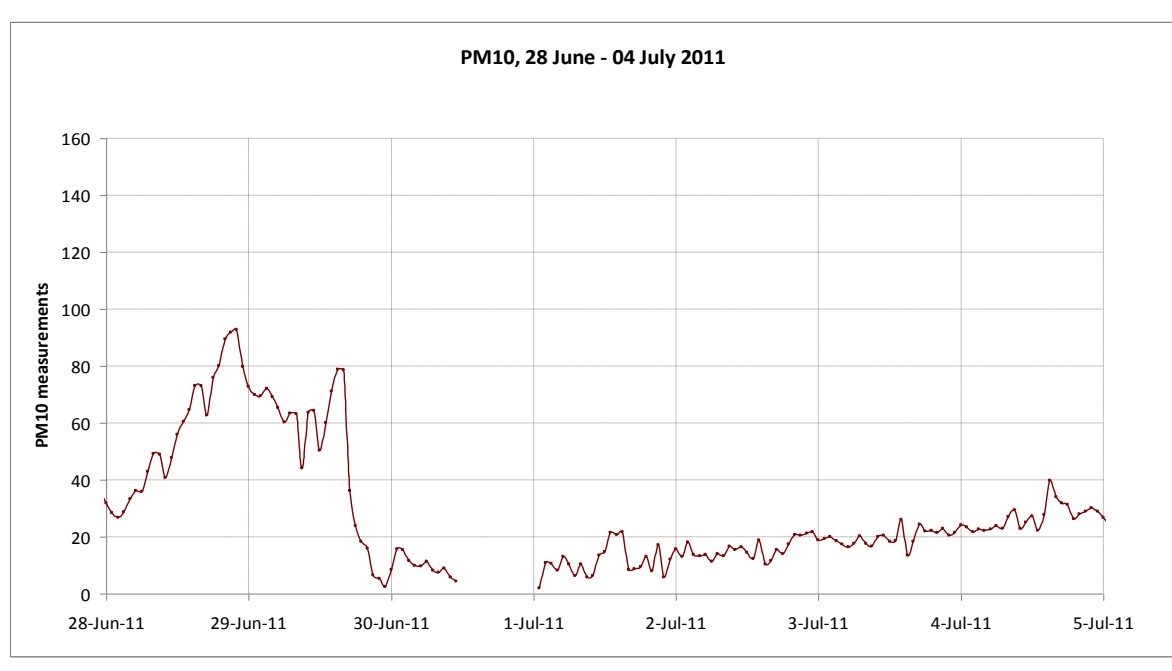

**Fig. 18b: Hourly PM$_{10}$ measurements (in µg m$^{-3}$) for 28 June-04 July 2011 at JRC-Ispra, Italy.**