# Peer review of "A study of the influence of tropospheric subsidence on spring and summer surface ozone concentrations at the JRC-Ispra station in northern Italy"

_Atmospheric Chemistry and Physics, 2019_

## Referee Comment (RC1) · Anonymous Referee #1 · 20 Jun 2019

This paper investigates the role of tropospheric subsidence on near-surface ozone concentrations of the JRC-Ispra monitoring station. The study is based on the analysis of several observational products (ground measurements, satellite retrievals), reanalysis data and back trajectories. This is an interesting and solid study, with adequate discussion of the results. However, I think that the structure and number of figures should be revised in order to be more reader-friendly. I recommend publication of the paper after the following comments are addressed.

Main comments

a. The manuscript includes too many figures. For example, the third examined episode is associated with 10 figures, while the first episode with only one. My suggestions on this are the following:

1. Merge Figure 1 and Figure 2 into one Figure. Increase the size of the legends in Figure 1 and Figure 2.

2. As Figure 6 and Figure 7 are referring to the same episode you can merge them into one Figure. The same stands for Figure 12 and Figure 13.

3. As Figures 16a and 16b are of the same temporal extent they can be combined into one figure with a secondary vertical axis. The same stands for Figure 18a and 18b. Then the new fig.16, fig. 17a, fig 17b, and the new fig18 can be merged into one Figure.

4. Please use a, b, c... labelling for all your Figures.

5. For every Figure use one caption describing there every a, b, c. . . subfigure.

6. In Figures 3-11, 14 and 15 remove the surrounding white space to improve both the quality and visibility of the figures.

b. Regarding the selection of the episodes the authors state that ". . .the 3 most characteristic of them.." will be presented (page 7, line 32). Most characteristic in terms of what? Can you be more specific on this somehow subjective criterion?

c. I suggest presenting the time period of each of the three episodes at the beginning of Section 3.2. Then every examined episode can be presented as individual Section 3.2.1, 3.2.2 and 3.2.3.

Comments

1. Apart from tropospheric subsidence influencing near-surface ozone concentrations, there are climatological and case studies of stratospheric intrusions affecting near-surface ozone concentrations for the Mediterranean region (Cristofanelli et al., 2006;

Gerasopoulos et al., 2006; Akritidis et al., 2010). I believe that the contribution of such events on near-surface ozone for the Mediterranean region should be also included in the Introduction.

2. Page 6, line 18: Please add a reference for the use of the 7Be/210Pb ratio and a small description for the purpose of its usage.

Technical comments

1. Page 2, line 7: Remove dot after "are observed."

2. Page 3, line 6: Replace "if 3.8 days" with "of 3.8 days".

3. Page 3, lines 12-13. Please correct the order of references. Also, check for other similar instances throughout the manuscript and correct accordingly.

4. Page 3, line 32: Replace "during summertime ozone episodes over the eastern Mediterranean and linked" with "during the summertime ozone episodes over the eastern Mediterranean and are linked".

5. Page 4, line 21 to Page 5 line 4. I suggest including bullets for the description of the measurements.

6. Page 4, line 22: Replace "Jensen et al., 2017" with "Jensen et al. (2017)". Also, check for other similar instances throughout the manuscript and correct accordingly.

7. Page 4, line 27: Delete the extra dot.

8. Page 5, line 6: Replace "charts for" with "charts of".

9. Page 5, line 7: Replace "for the atmospheric "with "at the atmospheric".

10. Page 6, line 17: Replace "and of ozone vs" with "and that of ozone vs".

11. Page 7, lines 31-32: Replace 10 and 3 with "Ten" and "three".

12. Page 11, lines 13-17. This is a rather long sentence. Please rephrase.

13. Page 32, Figure 10: Please rephrase the figure caption to be clearer.

References

Akritidis, D., Zanis, P., Pytharoulis, I., Mavrakis, A., and Karacostas, T.: A deep strato-spheric intrusion event down to the earth's surface of the megacity of Athens, Meteorol. Atmos. Phys., 109, 9–18, 2010

Cristofanelli, P., Bonasoni, P., Tositti, L., Bonafe, U., Calzolari, F., Evangelisti, F., San-drini, S., and Stohl, A.: A 6-year analysis of stratospheric intrusions and their influence on ozone at Mt. Cimone (2165m above sea level), J. Geophys. Res.-Atmos., 111, D03306, https://doi.org/10.1029/2005JD006553, 2006.

Gerasopoulos, E., Zanis, P., Papastefanou, C., Zerefos, C. S., Ioannidou, A., and Wernli, H.: A complex case study of down to the surface intrusions of persistent strato-spheric air over the Eastern Mediterranean, Atmos. Environ., 40, 4113–4125, 2006.

---

## Referee Comment (RC2) · Anonymous Referee #2 · 30 Sep 2019

The article presents specific case studies to interpret the role of subsidence on high elevation surface ozone concentrations, with the synergy of different in situ and satellite observational data. Overall, the paper merits publication after a number of comments are taken into account:

In the abstract, a 10 year measurement period is mentioned, which creates a clear expectation that the results of this study are put in a long term context. This is not happening and it is a clear gap of the study. Unless the criteria used to select the cases are analysed in a previous paper (in that case direct reference to result from that

paper(s) should be included), the authors should provide some statistics on the extent at which their findings for 2011 are also typical for the rest of the years as well as some means of quantification (e.g. frequency, values during events versus average values).

The amounts of plots used is huge! The authors should definitely make a serious attempt either to merge few of them, or move to suppl. material, or exclude if really not needed. Those changes might need to be followed by changes in the text and the overall structure, which I leave upon the authors, yet few suggestions will follow immediately after in my review.

Finally, there are parts of the introduction or the results where references do not seem to be up to date, either in terms of time or space, the latter meaning references relevant to the area of interest. I have included few examples which I consider only indicative, but a more through review of the current state might be needed, and the selection remains at the discrete consideration of the authors.

Specific comments:

Pg 1, Ln 30 – "It has been reported that tropospheric . . . the last couple of centuries (Volz and Kley, 1988; Forster et al., 2007)." I would suggest that this introductory statement should be supported with more recent references.

Pg2, Ln 6 – "which might also be associated to deep tropospheric subsidence especially over the Mediterranean . . . Kalabokas et al., 2013; Cooper et al., 2014; Safieddine et al., 2014; Kalabokas et al., 2015) . . . especially for deep stratospheric intrusions the following references are very characteristic for the area and should be included in the already too long list of references, or later (Pg 2, lines 20-25). • A deep stratospheric intrusion event down to the Earth's surface of the megacity of AthensApril 2012Meteorology and Atmospheric Physics 109(1):9-18, DOI: 10.1007/s00703-010-0096-6 by Akritidis et al. • Gerasopoulos E, Zanis P, Papastefanou C, Zerefos CS, Ioannidou A,Wernli H (2006) A complex case study of down to the surfaceintrusions of persistent stratospheric air over the EasternMediterranean. Atmos Environ 40:4113–

4125 • Kentarchos AS, Davies TD, Zerefos C (1998) A low latitude stratospheric intrusion associated with a cut-off low. GeophysRes Lett 25:67–70

Pg 2, Ln 31 – It seems that two studies conducted at Finokalia remote station in the eastern Mediterranean, dealing with the dynamics and photochemistry of ozone are missing from the introduction, especially when discussing the eastern Mediterranean controlling mechanisms of surface ozone. On the contrary, there are many self-citations from the first author that need to be enriched with studies from other groups in the area. • Kouvarakis, G., K. Tsigaridis, M. Kanakidou, and N. Mihalopoulos (2000), Temporal variations of surface regional background ozone over Crete Island in the southeast Mediterranean, J. Geophys. Res., 105(D4), 4399 – 4407. • Photochemical ozone production in the Eastern Mediterranean, June 2006, Atmospheric Environment 40(17):3057-3069, DOI: 10.1016/j.atmosenv.2005.12.061 by Gerasopouolos et al. • Gerasopoulos, E., G. Kouvarakis, M. Vrekoussis, M. Kanakidou, and N. Mihalopoulos (2005), Ozone variability in the marine boundary layer of the eastern Mediterranean based on 7-year observations, J. Geophys. Res., 110, D15309, doi:10.1029/2005JD005991

Pg 3, Ln 15-17: Be7 reference for ambient levels are quite old, some inquiry on new articles reporting on the levels should be done, especially in the area of interest. The same in lines 21-23.

The information in section 2.2 should be better included in a table.

Figure 1a could be combined with 1b. The same stands for 2a, 2b.

Overall, the added value of this paper results is not clear and should be better highlighted, mostly in the conclusions. It is obvious that it is an extension of previous works and for that reason it needs to be clear where does this study starts from and where it ends up (added value) at the same time being a self standing scientific publication.

---

## Author Comment (AC1) · 13 Nov 2019

**Anonymous Referee #1 Comments (Authors Response in Italics)**

This paper investigates the role of tropospheric subsidence on near-surface ozone concentrations of the JRC-Ispra monitoring station. The study is based on the analysis of several observational products (ground measurements, satellite retrievals), reanalysis data and back trajectories. This is an interesting and solid study, with adequate discussion of the results. However, I think that the structure and number of figures should be revised in order to be more reader-friendly. I recommend publication of the paper after the following comments are addressed.

*We would like to thank the reviewer for his positive comments on the manuscript. As seen below we revised the paper according to his suggestions.*
*The corresponding changes in the revised manuscript (in Final Response) are highlighted in yellowcolor.*

Main comments
a. The manuscript includes too many figures. For example, the third examined episode is associated with 10 figures, while the first episode with only one. My suggestions on this are the following:
1. Merge Figure 1 and Figure 2 into one Figure. Increase the size of the legends in Figure 1 and Figure 2.

*Figs 1 and 2 have been merged into one Figure (new Fig. 1) and the size of the legends has been increased.*

2. As Figure 6 and Figure 7 are referring to the same episode you can merge them into one Figure. The same stands for Figure 12 and Figure 13.

*Figs 6 and 7 have been merged into one Figure (new Fig. 5) as well as Figs 12 and 13(new Fig. 10)*

3. As Figures 16a and 16b are of the same temporal extent they can be combined into one figure with a secondary vertical axis. The same stands for Figure 18a and 18b. Then the new fig.16, fig. 17a, fig 17b, and the new fig18 can be merged into one Figure.

*Figs 16a, 16b, 17a, 17b, 18a, 18b, have been merged into one Figure (new Fig. 13)*

4. Please use a, b, c... labelling for all your Figures.
5. For every Figure use one caption describing there every a, b, c. . . subfigure.

*In the new Figures 1 and 13, presenting the JRC-station measurements, the labels a, b, c, d were used and also one caption was used describing the a, b, c, d subfigures.*

6. In Figures 3-11, 14 and 15 remove the surrounding white space to improve both the quality and visibility of the figures.

*For the indicated Figs the surrounding white space was removed.*

b. Regarding the selection of the episodes the authors state that ". . .the 3 most characteristic of them.." will be presented (page 7, line 32). Most characteristic in terms of what? Can you be more specific on this somehow subjective criterion?

*The sentence was rephrased as follows:*
*"More than ten 7Be - ozone weekly episodes were identified in the whole time series and the three most characteristic of them, for what concerns signs of tropospheric subsidence as observed in the meteorological and air pollution measurements (high 7Be and O3 concentrations combined with positive omega and dry air masses), will be presented in the following paragraphs. The selected episodes were: 3-10 May 2011, 23-29 May 2012 and 28 June – 04 July 2011".*

c. I suggest presenting the time period of each of the three episodes at the beginning of Section 3.2. Then every examined episode can be presented as individual Section 3.2.1, 3.2.2 and 3.2.3.

*Ok*

Comments
1. Apart from tropospheric subsidence influencing near-surface ozone concentrations, there are climatological and case studies of stratospheric intrusions affecting near surface ozone concentrations for the Mediterranean region (Cristofanelli et al., 2006; Gerasopoulos et al., 2006; Akritidis et al., 2010). I believe that the contribution of such events on near-surface ozone for the Mediterranean region should be also included in the Introduction.

*The contribution of stratospheric ozone intrusions affecting near surface ozone concentrations for the Mediterranean region has been more stressed in the introduction and the suggested relevant references have been included.*
*So, the following sentence was added:*
*Apart from tropospheric subsidence influencing near-surface ozone concentrations, It has to be mentioned also that there are climatological and case studies of stratospheric intrusions affecting near surface ozone concentrations for the Mediterranean region (Cristofanelli et al., 2006; Gerasopoulos et al., 2006; Akritidis et al., 2010)*

2. Page 6, line 18: Please add a reference for the use of the 7Be/210Pb ratio and a small description for the purpose of its usage.

*We think that this information is already presented sufficiently in the introduction (Page 3, lines 1-24), where many relevant publications on the use of radionuclide measurements for atmospheric transport are cited. We added also a recent relevant paper, which describes very extensively the use of the 7Be and 210Pb radionuclides in atmospheric transport studiesin its introduction (Brattich et al., 2017). The referenced WMO-GAW report (2004) as well as the cited reference Koch et al., 1996 make also an interesting review on this subject.*
*In addition, the following phrase was added in the 1ˢᵗ line of the respective paragraph in the introduction:*
*"and in particular terrigenous 210Pb and cosmogenic 7Be, which are natural radionuclides that are helpful in understanding the roles of transport and/or scavenging in controlling the behavior of radiatively active trace gases and aerosol"*

Technical comments
1. Page 2, line 7: Remove dot after "are observed."
*Ok*

2. Page 3, line 6: Replace "if 3.8 days" with "of 3.8 days".
*Ok*

3. Page 3, lines 12-13. Please correct the order of references. Also, check for other similar instances throughout the manuscript and correct accordingly.
*Ok*

4. Page 3, line 32: Replace "during summertime ozone episodes over the eastern Mediterranean and linked" with "during the summertime ozone episodes over the eastern Mediterranean and are linked".
*Ok*

5. Page 4, line 21 to Page 5 line 4. I suggest including bullets for the description of the measurements.
*Ok*

6. Page 4, line 22: Replace "Jensen et al., 2017" with "Jensen et al. (2017)". Also, check for other similar instances throughout the manuscript and correct accordingly.
*Ok*

7. Page 4, line 27: Delete the extra dot.
*Ok*

8. Page 5, line 6: Replace "charts for" with "charts of".
*Ok*

9. Page 5, line 7: Replace "for the atmospheric "with "at the atmospheric".
*Ok*

10. Page 6, line 17: Replace "and of ozone vs" with "and that of ozone vs".
*Ok*

11. Page 7, lines 31-32: Replace 10 and 3 with "Ten" and "three".
*Ok*

12. Page 11, lines 13-17. This is a rather long sentence. Please rephrase.

*The sentence was modified as follows:*
*"During high 7Be and high ozone episodes, the highest evening ozone values exceeding the standards usually occur within the following 2-3 days after the maximum of regional tropospheric subsidence, as observed also in the analysis of several episodes not presented in this paper. The increase in ozone concentrations usually occurs under the influence of favourable meteorological conditions for photochemical ozone production in the boundary layer, which is added-up on the increased regional background due to tropospheric subsidence and thus occasionally leading to exceedances in ozone air quality standards".*

13. Page 32, Figure 10: Please rephrase the figure caption to be clearer.

*The figure caption was rephrased.*

References
Akritidis, D., Zanis, P., Pytharoulis, I., Mavrakis, A., and Karacostas, T.: A deep stratospheric intrusion event down to the earth's surface of the megacity of Athens, Meteorol. Atmos. Phys., 109, 9–18, 2010
Cristofanelli, P., Bonasoni, P., Tositti, L., Bonafe, U., Calzolari, F., Evangelisti, F., Sandrini, S., and Stohl, A.: A 6-year analysis of stratospheric intrusions and their influence on ozone at Mt. Cimone (2165m above sea level), J. Geophys. Res.-Atmos., 111, D03306, https://doi.org/10.1029/2005JD006553, 2006.
Gerasopoulos, E., Zanis, P., Papastefanou, C., Zerefos, C. S., Ioannidou, A., and Wernli, H.: A complex case study of down to the surface intrusions of persistent stratospheric air over the Eastern Mediterranean, Atmos. Environ., 40, 4113–4125, 2006.

---

## Author Comment (AC2) · 13 Nov 2019

**Anonymous Referee #2 Comments (Authors Response in Italics)**

The article presents specific case studies to interpret the role of subsidence on high elevation surface ozone concentrations, with the synergy of different in situ and satellite observational data. Overall, the paper merits publication after a number of comments are taken into account:

*We would like to thank the reviewer for his positive comments on the manuscript. As seen below we revised the paper according to his suggestions.*
*The corresponding changes in the revised manuscript (in Final Response) are highlighted in green color.*

In the abstract, a 10 year measurement period is mentioned, which creates a clear expectation that the results of this study are put in a long term context. This is not happening and it is a clear gap of the study. Unless the criteria used to select the cases are analysed in a previous paper (in that case direct reference to result from that paper(s) should be included), the authors should provide some statistics on the extent at which their findings for 2011 are also typical for the rest of the years as well as some means of quantification (e.g. frequency, values during events versus average values).

*We would like to emphasize that the study is focused on atmospheric mechanisms, based on some selected case-studies and it is not a statistical one. In combination with a similar comment of reviewer 1 the relevant phrase (page 8, lines 3-7) was modified as follows "More than ten 7Be - ozone weekly episodes were identified in the whole time series and the three most characteristic of them, for what concerns signs of tropospheric subsidence as observed in the meteorological and air pollution measurements (high 7Be  and O3 concentrations combined with positive omega and dry air masses) will be presented and examined in the following paragraphs. The selected episodes were: 3-10 May 2011, 23-29 May 2012 and 28 June – 04 July 2011. The episodes discussed here are not Foehn events".*
*In addition, the reader might  have an idea of the frequency of occurrence of these episodes by having a look at Fig. 1 as well as in the Supplement Figs S1-S4 where the weekly averages of 7Be and O3 concentrations for 5 years (2006, 2007, 2008, 2011, 2012) plotted. As observed, during the April – September period about 2-3 major 7Be - O3 episodes are spotted. As mentioned in the manuscript, these episodes could be better detected with shorter than weekly measurements of 7Be as the usual duration of subsidence episodes is about 2-3 days (see new Fig. 13), but such measurements were not available.*

The amounts of plots used is huge! The authors should definitely make a serious attempt either to merge few of them, or move to suppl. material, or exclude if really not needed. Those changes might need to be followed by changes in the text and the overall structure, which I leave upon the authors, yet few suggestions will follow immediately after in my review.

*A significant merging and reduction of plots associated with the corresponding text changes have been undertaken in combination also with similar comments of reviewer 1, leading to a total number of 13 Figures in the revised text.*

Finally, there are parts of the introduction or the results where references do not seem to be up to date, either in terms of time or space, the latter meaning references relevant to the area of interest. I have included few examples which I consider only indicative, but a more through review of the current state might be needed, and the selection remains at the discrete consideration of the authors.

*Following the reviewer's comment an update of references has been made in the introduction and the results section, as it is described in the following paragraphs.*

Specific comments:
Pg 1, Ln 30 – "It has been reported that tropospheric . . . the last couple of centuries (Volz and Kley, 1988; Forster et al., 2007)." I would suggest that this introductory statement should be supported with more recent references.

*Following the reviewer's suggestion two more recent references, which are review papers on tropospheric ozone were mentioned for supporting the introductory statement (Monks et al., 2015; Gaudel et al., 2018).*

Pg2, Ln 6 – "which might also be associated to deep tropospheric subsidence especially over the Mediterranean . . . Kalabokas et al., 2013; Cooper et al., 2014; Safieddine et al., 2014; Kalabokas et al., 2015) . . . especially for deep stratospheric intrusions the following references are very characteristic for the area and should be included in the already too long list of references, or later (Pg 2, lines 20-25). A deep stratospheric intrusion event down to the Earth's surface of the megacity of Athens April 2012 Meteorology and Atmospheric Physics 109(1):9-18, DOI: 10.1007/s00703-010-0096-6 by Akritidis et al. Gerasopoulos E, Zanis P, Papastefanou C, Zerefos CS, Ioannidou A,Wernli H (2006) A complex case study of down to the surface intrusions of persistent stratospheric air over the EasternMediterranean. Atmos Environ 40:4113–4125. Kentarchos AS, Davies TD, Zerefos C (1998) A low latitude stratospheric intrusion associated with a cut-off low. Geophys. Res. Lett. 25:67–70

*Following the reviewer's suggestion, the mentionned references on deep tropospheric subsidence over the Mediterranean were added into the text.*

Pg 2, Ln 31 – It seems that two studies conducted at Finokalia remote station in the eastern Mediterranean, dealing with the dynamics and photochemistry of ozone are missing from the introduction, especially when discussing the eastern Mediterranean controlling mechanisms of surface ozone. On the contrary, there are many selfcitations from the first author that need to be enriched with studies from other groups in the area. Kouvarakis, G., K. Tsigaridis, M. Kanakidou, and N. Mihalopoulos (2000), Temporal variations of surface regional background ozone over Crete Island in the southeast Mediterranean, J. Geophys. Res., 105(D4), 4399 – 4407. Photochemical ozone production in the Eastern Mediterranean, June 2006, Atmospheric Environment 40(17):3057-3069, DOI: 10.1016/j.atmosenv.2005.12.061 by Gerasopouolos et al. Gerasopoulos, E., G. Kouvarakis, M. Vrekoussis, M. Kanakidou, and N. Mihalopoulos (2005), Ozone variability in the marine boundary layer of the eastern Mediterranean based on 7-year observations, J. Geophys. Res., 110, D15309, doi:10.1029/2005JD005991.

*The suggested studies on surface ozone conducted at the Finokalia station were added into the manuscript.*

Pg 3, Ln 15-17: Be7 reference for ambient levels are quite old, some inquiry on new articles reporting on the levels should be done, especially in the area of interest. The same in lines 21-23.

*Following the reviewer's suggestion ten more recent references on 7Be ambient levels were added, especially concerning the area of interest of the study (European Continent – Western Mediterranean): Bourcier et al., 2011; Brattich et al 2017; Duenas et al, 2011; García et al, 2012; Hernández-Ceballos et al, 2016; Ioannidou et al, 2014; Jiwen et al, 2013; Leppanen et al, 2010; Lozano et al, 2012; Pham et al, 2011; Steinmann et al, 2013.*

The information in section 2.2 should be better included in a table.

*A table has been added to include the information in section 2.2 (Instrumentations and measurements at JRC-Ispra site).*

Figure 1a could be combined with 1b. The same stands for 2a, 2b.

*The suggested combination of Figs has been done following also a similar comment from reviewer 1.*

Overall, the added value of this paper results is not clear and should be better highlighted, mostly in the conclusions. It is obvious that it is an extension of previous works and for that reason it needs to be clear where does this study starts from and where it ends up (added value) at the same time being a self standing scientific publication.

*In the previous study (Kalabokas et al., 2017) some troposheric mechanisms related with regional ozone episodes especially linked with large-scale subsidence were examined mainly based on surface ozone, IASI vertical columns and meteorological analysis. In this manuscript a more detailed analysis of the suggested mechanisms was performed, based on the measurements of a very large variety of meteorological and air pollution parameters collected at the JRC-Ispra station, which is considered as one of the most well-equipped measuring sites in Europe. This measurement set includes tracers of both subsidence ($^7$Be, RH), boundary layer origin ($^{222}$Rn, $^{210}$Pb, NOx) and photochemical activity (partly PM), and this allows at least qualitatively distinguish origin of different air masses and trace back ozone origin. Relevant phrases have been inserted at the last paragraph of the introduction as well as at the beginning of the conclusions section.*

---

## Author Comment (AC3) · 13 Nov 2019

The comment was uploaded in the form of a supplement:
https://www.atmos-chem-phys-discuss.net/acp-2019-438/acp-2019-438-AC3-supplement.pdf